# Linkages between maternal experience of intimate partner violence and child nutrition outcomes: A rapid evidence assessment

Silvia Bhatt Carreno[1], Manuela Orjuela-Grimm[2], Luissa Vahedi[3], Elisabeth Roesch[4], Christine Heckman[4], Andrew Beckingham[4], Megan Gayford[4], Sarah R. Meyer[5]*

1 Department of Epidemiology, Columbia University, New York City, New York, United States of America, 2 Department of Epidemiology and Pediatrics, Columbia University Irving Medical Center, New York City, New York, United States of America, 3 Brown School, Washington University in St. Louis, St. Louis, Missouri, United States of America, 4 UNICEF, New York City, New York, United States of America, 5 Institute for Medical Information Processing, Biometry, and Epidemiology, Ludwig-Maximilians-Universität, Munich, Germany

* Sarah.meyer@ibe.med.uni-muenchen.de

## Abstract

### Background

A strong evidence base indicates that maternal caregivers' experience of intimate partner violence [IPV] impacts children's health, cognitive development, and risk-taking behaviors. Our objective was to review peer-reviewed literature describing the associations between a child's indirect exposure to IPV and corresponding nutrition outcomes, with a particular focus on fragile settings in low and middle-income countries [LMICs].

### Methods

We conducted a rapid evidence assessment to synthesize quantitative associations between maternal caregivers' IPV experience and children's nutrition/growth outcomes (birthweight, feeding, and growth indicators). We included peer-reviewed research, published in English or Spanish after the year 2000, conducted in fragile settings in LMICs.

### Results

We identified 86 publications that fit inclusion criteria. Amongst all associations assessed, a maternal caregiver's experience of combined forms of IPV (physical, sexual and emotional) or physical IPV only, were most consistently associated with lower birthweight, especially during pregnancy. Women of child-bearing age, including adolescents, exposed to at least one type of IPV showed a decreased likelihood of following recommended breastfeeding practices. Lifetime maternal experience of combined IPV was significantly associated with stunting among children under 5 years of age in the largest study included, though findings in smaller studies were inconsistent. Maternal experience of physical or combined IPV were inconsistently associated with underweight or wasting in the first five years. Maternal experience of sexual IPV during pregnancy appeared to predict worsened lipid profiles among children.

**Data Availability Statement:** All relevant data are within the manuscript and its Supporting Information files.

**Funding:** The rapid evidence assessment was funded by the Safe from the Start grant to UNICEF, from Bureau of Population and Migration The funders had no role in study design, data collection and analysis, decision to publish, or preparation of the manuscript.

**Competing interests:** The authors have declared that no competing interests exist.

## Conclusion

Maternal caregivers' experience of IPV is significantly associated with low birthweight and suboptimal breastfeeding practices, whereas studies showed inconsistent associations with child growth indicators or blood nutrient levels. Future research should focus on outcomes in children aged 2 years and older, investigation of feeding practices beyond breastfeeding, and examination of risk during time periods physiologically relevant to the outcomes. Programmatic implications include incorporation of GBV considerations into nutrition policies and programming and integrating GBV prevention and response into mother and child health and nutrition interventions in LMIC contexts.

## Introduction

Deficiencies in children's growth and nutrition are multifactorial in origin, but potential contributors include social inequalities in food distribution [1] and gender inequality [2]. Intimate partner violence [IPV], defined as "behavior by an intimate partner or ex-partner that causes physical, sexual or psychological harm, including physical aggression, sexual coercion, psychological abuse and controlling behaviors" [3] may also influence children's nutrition outcomes. Different types of IPV are physical IPV (which includes slapping, hitting, kicking and beating [4]), sexual IPV (which includes forced sexual intercourse and other forms of sexual coercion, whether using physical force or other forms of coercion) [4], emotional or psychological IPV (which includes insults, belittling, humiliation, and intimidation [4]), and controlling behaviors (which includes actions such as a partner limiting respondent's contact with family or friends or monitoring and restricting money management, movement, education, employment or medical care) [5, 6]. Male perpetration of IPV against women and girls is highly prevalent globally [7], and in fragile settings and humanitarian emergencies, the risk factors increase [8]. A 2011 systematic review of population-based studies focused on prevalence of all forms of violence against women and girls in complex emergencies indicated that IPV is one of the most common forms of violence experienced by women and girls in humanitarian contexts [9]; limited evidence confirms that IPV prevalence increases in conflict-affected areas compared to non-conflict settings [10].

Children–defined as anyone from birth until age 18, as per international definitions–can be indirectly exposed to IPV, which can include witnessing, being aware of, or secondarily affected by the presence of violence against a maternal caregiver in the household. Children's indirect exposure to IPV is associated with lower rates of vaccination and with later adolescent and adult risk-taking behaviors [11]. There is robust literature on indirect IPV exposure and children's adverse mental health outcomes, such as externalizing symptoms, (e.g. aggression, impulsivity, or anti-social behavior) [11]. Evidence indicates that pre-school aged children who witness IPV in childhood may experience detrimental effects on their later mental, social and physical health during adolescence and adulthood. Some children develop traumatic stress symptoms, mood anxiety disorders, depression, alcohol use, and have higher rates of violence perpetration [12–15]. A recent review of IPV against a maternal caregiver and child development outcomes in 11 low- and middle-income countries [LMICs] identified associations between IPV exposure and literacy, numeracy, and cognitive, physical and socioemotional development for children aged 36 to 59 months [16]. For example, a study conducted in South Africa showed that emotional IPV was associated with lower language development, motor

development and cognitive scores in children at age 2, while physical IPV was associated with lower motor scores [17], and in a study conducted in Kenya, maternal exposure to IPV was associated with poorer child behavioral outcomes [18]. Girls and boys may experience different patterns of direct and indirect exposure to IPV, however, the limited research exploring these differences is mixed [19, 20], and there are also gender differences in mental health impacts of childhood IPV exposure [21]. The evidence-base concerning the impacts of childhood indirect exposure to IPV on children's health outcomes is outside of mental health disorders and risk-taking behaviors is limited.

Nutritional vulnerabilities pose particular challenges in countries defined as 'fragile or extremely fragile,' which can be defined as countries "where the state power is unable and/or unwilling to deliver core functions to the majority of its people: security, protection of property, basic public services and essential infrastructure" [22]. Countries where there is national and/or regional fragility are more likely to have high levels of anemia, and those countries deemed as extremely fragile are more likely to have high levels of anemia as well as high levels of stunting and wasting (both markers of chronic malnutrition) [23, 24]. For example, of the estimated 45.4 million children under 5 globally who suffer from wasting, one in four live in humanitarian contexts [24]. Existing evidence gaps illustrate that nutrition indicators are especially challenging to study and address in fragile settings. Indeed, Yount et al.'s review, published a decade ago, focused on presenting a possible framework for pathways between IPV and child growth and nutrition in the first 36 months, primarily included studies based on data gathered in HIC settings, and, to a much lesser degree, LMIC and "fragile" settings [25].

Several evidence syntheses have addressed the question of the association between children's indirect exposure to IPV (against a maternal caregiver) and child nutrition outcomes. Yount et al.'s review found that IPV "may affect early childhood growth and nutrition through biological and behavioral pathways," and that the strongest evidence available concerned the association between IPV and low birth weight [LBW], an association this review indicated is likely mediated by maternal prenatal risk behaviors, mental health and poor weight gain during pregnancy [25]. Other relevant reviews have focused on in-utero exposure to IPV rather maternal experience of IPV at other time points; Hill et al. found associations between IPV during pregnancy and LBW and preterm birth [26] and Donovan et al. found associations between IPV during pregnancy and pre-term birth, LBW and small for gestational age babies [27]. Experience of IPV can influence maternal care practices relating to child nutrition; a review of observational studies found that 8 of 12 included studies reported an association between IPV and breastfeeding practices (lower breastfeeding intention, breastfeeding initiation and exclusive breastfeeding) [28]. These existing evidence analyses focus on specific periods of experiencing IPV (in-utero), or a limited range of nutrition indicators (LBW, breastfeeding). To address existing evidence gaps, we conducted a rapid evidence assessment [REA] focusing on all forms of indirect experience of IPV, during any exposure period, and including nutritional/growth outcomes reported as a) fetal growth (measured indirectly with birth weight), b) infant feeding practices, as well as indicators of c) child growth (using anthropometry) and d) nutrient blood markers in order to examine acute and chronic malnutrition during infancy, early childhood (0 to 39 months, including those focused on children under 2 or under 5 years), and middle childhood (after the early growth spurt). Examining pathways concerning children's indirect exposure to IPV against a maternal caregiver and childhood nutrition and growth can highlight how IPV adversely impacts child health and inform policies and programs at the nexus of IPV and nutrition. Our objective was to review and synthesize the current quantitative literature describing the associations between IPV against a maternal caregiver and children's nutrition outcomes, with a particular focus on those studies carried out in LMICs and fragile settings.

## Methods

The REA was conducted utilizing an adapted systematic review methodology [29]. Specifically, title/ abstract review was conducted by a single reviewer, and inclusion/ exclusion criteria (specifically, regarding the location of studies for inclusion) was determined iteratively. We developed and implemented a structured search of three databases: Medline, Embase, and Global Health. The initial search was conducted in July 2021, and updated in October 2022. The full search strategy is included in S1 Appendix. For further details on search methodology, see [blinded for peer review] et al. In brief, the search terms included the following fields: i) intimate partner violence, ii) nutritional outcomes, and iii) quantitative study design, with specific terms and MeSH headings tailored to each database.

The eligibility criteria were structured using the PECO framework–Population, Exposure, Context and Outcome [30] (see Table 1). The timepoints considered were during infancy, early childhood (including those outcomes specific to children under 2 or under 5 years), middle childhood and adolescence (after puberty or above age 10 years). Other inclusion criteria were that the manuscript was published in peer-reviewed literature, published in English or Spanish, after the year 2000. Spanish was chosen as a second language for the review due to the language capacity of the team. Any quantitative methodology studies were included, along with mixed methods studies if quantitative findings were separately reported.

All records identified through the database searches were downloaded to Covidence, a systematic review software. Screening occurred in two stages: (i) title and abstract; and (ii) full text review. Title/ abstracts were each screened by one reviewer [one of SBC, MOG, LV or SRM]. During the full text screening, sources were assessed independently by two reviewers vis-à-vis the inclusion/ exclusion criteria [two of SBC, MOG, LV or SRM]. A third independent reviewer resolved any discord between the first two reviewers.

### Data extraction and assessment of quality of included studies

We extracted data pertaining to general study characteristics, study design and sampling, measurement of experience of IPV as well as nutrition outcomes, and summary of results (see Supplementary Materials for data extraction materials). The data extraction was completed by a single reviewer [one of SBC or SRM] and checked for consistency and accuracy by a second reviewer [MOG or SRM]. The quality assessment measure included a series of questions that aim to assess methodological rigor including sources of poor measurement, bias, and degree to which results are externally valid. Due to the variability in study designs the team decided to use questions from four quality assessment tools: Mixed Methods Assessment Tool [MMAT] [31], Appraisal Tool for Cross Sectional Studies (AXIS) [32], Newcastle-Ottawa Scale (NOS) [33], and the National Institutes of Health assessment tool for observational cohort and cross sectional studies [34]. Specifically, the items whether the research objective was clearly stated, valid measurement of exposure (IPV) and outcome (nutrition) variables, and inclusion of relevant confounders were drawn from the NIH instrument, external validity (representativeness

**Table 1. Inclusion criteria.**

| Population | Boys or girls, under the age of 18 |
|---|---|
| Exposure | Any indirect exposure to IPV (in-utero and/ or household IPV exposures). IPV included physical, sexual or psychological violence or controlling behaviors, perpetrated against a maternal caregiver. |
| Context | A country in receipt of United Nations Central Emergency Response Funding any time during 2006–2021 |
| Outcome | a) fetal growth (measured indirectly with birth weight), b) breastfeeding, including infant feeding practices, c) indicators of child growth (using anthropometry), and d) nutrient blood markers. |

of sampling) and internal validity (non-response rates) items were drawn from the NOS instrument, an item about inclusion and exclusion criteria is drawn from the AXIS tool, and an item focused on sampling strategy is drawn from MMAT. This combination of items best fit the review objectives and included studies; previous systematic reviews have similarly combined quality assessment instruments [35].

### Data synthesis

Following data extraction and quality assessment, the team reviewed the included studies for patterns and summarized findings using tables and aggregate descriptive statistics (count frequencies and percentages). Data synthesis also included developing harvest plots to visually represent the associations identified in studies; the method was adapted to the purposes of this review, whereby associations were not weighted by study quality [36, 37].

## Results

There were 6493 sources imported to Covidence. Duplicates (1516) were removed, yielding 4977 citations for screening. During title and abstract screening, 4461 sources were excluded and 468 full texts were assessed for eligibility. During full text review, 450 sources were excluded for various reasons outlined in Fig 1. Specific characteristics of the 86 articles included are listed in Table 2. There were 57 cross-sectional studies, 23 prospective cohort studies, 5 case-control studies, one observational cohort study, and one prospective case-control study. About 50% of the included studies were published in the past five years, with 20 studies published in 2021 or 2022, indicating a rapidly expanding evidence-base. Study results are in displayed in S1 Table.

Table 3 displays results from the quality assessment. All individual quality assessment results are included in S2 Table. The sampling approach was highly variable, with 39.5% utilizing probability-based sampling methods and just over half (53.5%) including convenience sampling of women (typically women attending prenatal care or women who had given birth in facility/ hospital). Studies that utilized probability-based sampling methods were secondary analyses of Demographic and Health Surveys [DHS] data. In 40.7% of studies had a response rate of more than 90%. In 50.0% (n = 43) of studies, there was no description of the response rate and/ or any differences in characteristics between respondents and non-respondents. Despite these limitations, a high proportion of studies were categorized by the review team as having utilized rigorous measurement of IPV, i.e. questions that address acts-based items, rather than general questions about IPV experience (91.9%), and a similarly high proportion (87.2%) of studies included a rigorous measure of the nutrition outcomes (e.g. Z scores).

The REA findings are reported here by child developmental stages in the following sections: Low Birth Weight, Feeding Practices including Breastfeeding, and Indicators of Undernutrition (i.e. stunting, or wasting/underweight). The associations between maternal experience of IPV (lifetime, past year, and during pregnancy, wherein the affected child was in utero) and child nutrition and growth outcomes are shown the harvest plot in Fig 2. IPV was assessed separately (i.e. physical IPV only, sexual IPV only, emotional/ psychological/ verbal IPV only and controlling behaviors only) and as combined IPV, defined as physical, sexual and emotional, physical and sexual, or combined IPV–physical, sexual and emotional, physical and sexual, or any combination of IPV types. For each type of outcome, we include details from select studies that illustrate the associations found.

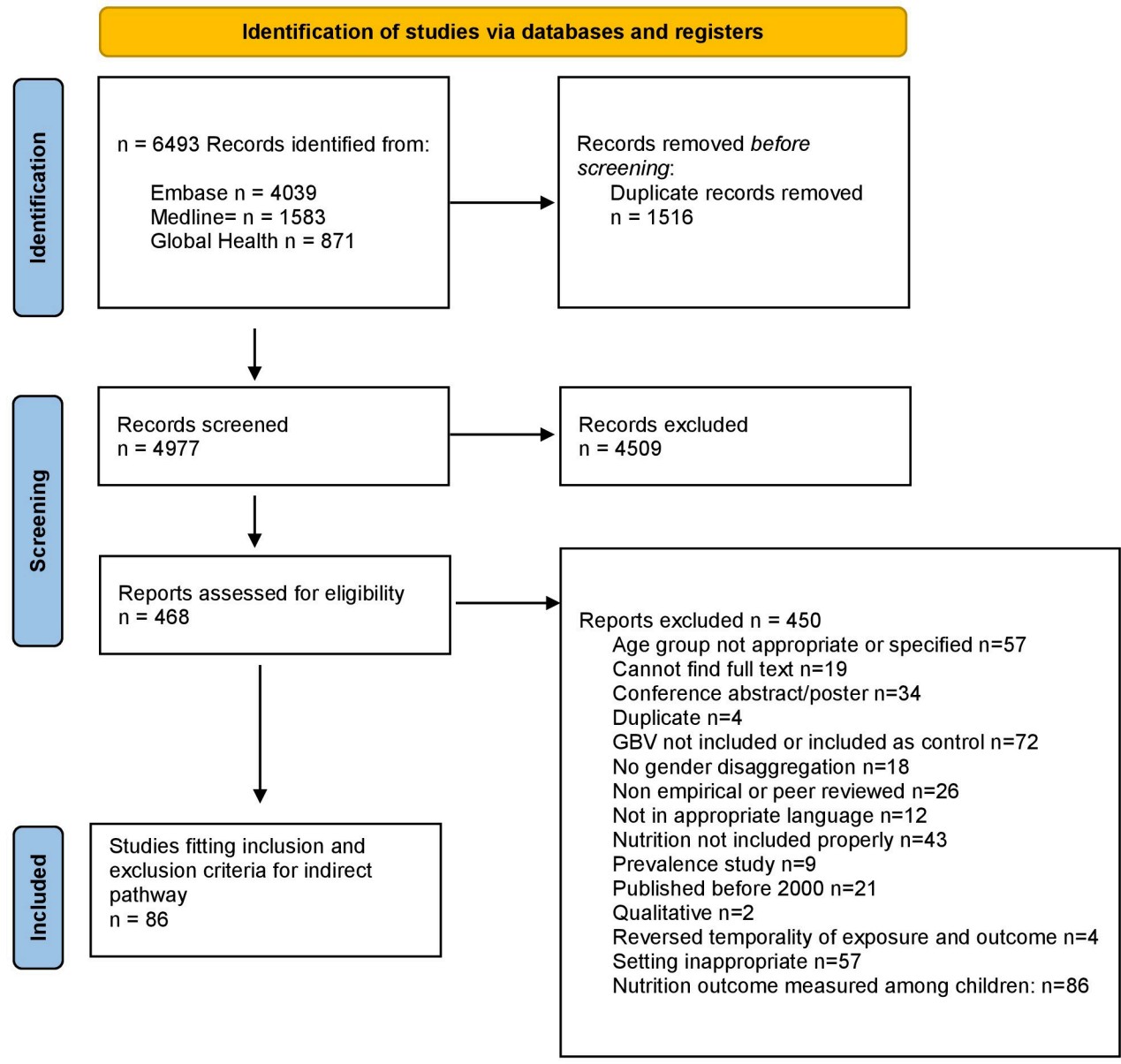

**Fig 1. PRISMA flow chart.**

### Associations with low birth weight (LBW)

Fig 2A shows the associations found between lifetime experience of IPV, experience of IPV in the past year, or experience of IPV during pregnancy, and LBW. One study did not specify a time period for IPV experience and indicated a positive association [38].

A relationship between lifetime experience of any type of IPV and LBW was examined in twelve studies. Higher experience of lifetime physical IPV was significantly associated with LBW in all four studies that examined this association [39–42]. Positive associations were also found in six of seven studies examining lifetime combined IPV and LBW [39–45]. Of the two studies that looked at lifetime sexual IPV and LBW, one found a significant association [39, 40]. Of the two studies that looked at lifetime emotional/psychological/verbal IPV and LBW, one found a significant association [39, 41].

**Table 2. Study characteristics–Indirect pathway.**

| Article | Research Question | Country, Region | Study design | Used secondary data Yes/ no If yes, which dataset? |
|---|---|---|---|---|
| Frith (2017) | To investigate the effect of breastfeeding counseling on the association between DV and EBF duration. | Bangladesh, SEAR | Longitudinal | No |
| Khan (2020) | To explore the impact of adverse maternal circumstances on LBW. | Bangladesh, SEAR | Cross sectional | Yes–DHS |
| Neamah (2018) | To explore how maternal experiences of IPV and depression impact a child's development and nutrition outcomes. | Tanzania, AFR | Cross sectional | No |
| Vo (2019) | To examine the association between DV during pregnancy and pre-term birth or LBW. | Vietnam, WPR | Cross sectional | No |
| Tran (2020) | To investigate how postpartum IPV and IPV experienced during pregnancy influences maternal mental health outcomes and breastfeeding practices. | Bangladesh, SEAR | Cross sectional | No |
| Madsen (2019) | To examine if IPV affects early termination of EBF. | Tanzania, AFR | Prospective cohort | No |
| Salazar (2012) | To examine association between exposure to IPV during pregnancy and linear growth of the child. | Nicaragua, AMR | Prospective cohort | No |
| Caleyachetty (2019) | To investigate how exposure to IPV affects initiation of early breastfeeding and EBF. | Afghanistan, Angola, Armenia, Azerbaijan, Bangladesh, Burkina Faso, Burundi, Cambodia, Cameroon, Chad, Colombia, Comoros, Cote d'Ivoire, Democratic Republic of Congo, Dominican Republic, Egypt, Ethiopia, Gabon, Ghana, Guatemala, Haiti, Honduras, India, Jordan, Kenya, Kyrgyz Republic, Liberia, Malawi, Maldives, Mali, Moldova, Mozambique, Myanmar, Namibia, Nepal, Nigeria, Pakistan, Peru, Philippines, Rwanda, Sao Tome and Principe, Sierra Leone, South Africa, Tajikistan, Tanzania, Timor Leste, Togo, Uganda, Ukraine, Zambia, Zimbabwe AFR, AMR, EMR, EUR, SEAR, WPR | Cross sectional | Yes–DHS |
| Hoang (2016) | To examine how exposure to IPV impacts the risk of adverse birth outcomes. | Vietnam, WPR | Prospective cohort | No |
| Kana (2020) | To analyze the association between exposure to IPV and birth weight. | Nigeria, AFR | Cross sectional | No |
| Rahman (2012) | To explore the association between physical and sexual IPV and nutritional outcomes such as being underweight, stunting and wasting. | Bangladesh, SEAR | Cross sectional | Yes–DHS |
| Ariyo (2021) | To examine the association between pregnancy, postpartum IPV and EBF. | Nigeria, AFR | Cross sectional | Yes–DHS |
| Shirin Ziaei (2014) | To investigate the association between ever married women experiencing IPV and nutrition outcomes in their children under five years old. | Bangladesh, SEAR | Cross sectional | Yes–DHS |
| Ziaei (2019) | To investigate how maternal exposure to DV impacts child lipid biomarkers. | Bangladesh, SEAR | Longitudinal | No |
| Ferdos (2017) | To explore the association between experience of maternal IPV and LBW. | Bangladesh, SEAR | Cross sectional | No |
| Batool (2018) | To explore the impact of IPV on child mortality and health in Pakistan. | Pakistan, EMR | Cross sectional | Yes–DHS |

(*Continued*)

**Table 2.** (Continued)

| Article | Research Question | Country, Region | Study design | Used secondary data Yes/ no If yes, which dataset? |
|---|---|---|---|---|
| Asling-Monemi (2009) | To explore the association between women's exposure to violence, the risk of fetal and early childhood malnutrition and the risk of under-five mortality. | Bangladesh, SEAR | Longitudinal | Yes–Prenatal food and micronutrient supplementation trial |
| Valladares (2002) | To explore the association between experiences of maternal physical violence and LBW in infants. | Nicaragua, AMR | Case Control | No |
| Valladares (2009) | To study the neuroendocrine release of cortisol in response to perceived stress among pregnant women exposed to partner violence and how this affects the duration of pregnancy and the intrauterine growth of the infant. | Nicaragua, AMR | Cross-sectional | No |
| Rahman (2021) | To explore how different types of IPV can affect birth outcomes such as LBW. | India, SEAR | Cross sectional | Yes–DHS National Family Health Survey (NFHS)-4 |
| Marimuthu (2019) | To explore the association between spousal support/abuse in pregnancy and LBW. | India, SEAR | Case Control | No |
| Tiwari (2018) | To explore the relationship between exposure to lifetime partner emotional abuse and reproductive outcomes and behaviors in Indian women. | India, SEAR | Cross sectional | Yes–DHS NFHS-3 |
| Sabu (2020) | To assess inequality of using the Composite index of anthropometric failure (CIAF) between tribal communities in Kerala and identify individual, parental and household factors affecting child undernutrition. | India, SEAR | Cross sectional | No |
| Subramanian (2008) | To explore the association between DV and malnutrition. | India, SEAR | Cross sectional | Yes–DHS NFHS-3 |
| Boyce (2017) | To explore the relationship between IPV and postnatal health practices with the goal of helping guide interventions promoting neonatal survival. | India, SEAR | Cross sectional | Yes–Data from Ananya program created to increase maternal and child health care utilization |
| Young (2020) | To examine multiple influences such as maternal, household, community and health service, on breastfeeding. | India, SEAR | Cross sectional | Yes–Data from a household survey done in 2017 collected for a maternal nutrition programme evaluation study |
| Zureick-Brown (2015) | To examine the association between physical or sexual IPV and feeding practices for newborns and infants within 24 hrs of birth. | India, SEAR | Cross sectional | Yes–DHS NFHS |
| Misch (2014) | To assess the impact of physical, sexual and/ or emotional maternal IPV victimization on early and exclusive breastfeeding practices in infants in different African countries. | Ghana, Kenya, Liberia, Malawi, Nigeria, Tanzania, Zambia, Zimbabwe, AFR | Cross sectional | Yes–DHS |
| Kaye (2006) | To determine the relationship between DV and LBW/maternal ill health among pregnant women. | Uganda, AFR | Prospective cohort | No |
| Sigalla (2017) | To investigate the relationship between IPV during pregnancy, preterm birth and LBW in Tanzania. | Tanzania, AFR | Prospective cohort | No |
| Musa (2021) | To examine the relationship between LBW, preterm birth and IPV during pregnancy. | Ethiopia, AFR | Cross sectional | No |
| Tsedal (2021) | To investigate the relationship between maternal IPV and diet among their children aged 6–23 months. | Ethiopia, AFR | Cross sectional | Yes–DHS |

(*Continued*)

**Table 2.** (Continued)

| Article | Research Question | Country, Region | Study design | Used secondary data Yes/ no If yes, which dataset? |
|---|---|---|---|---|
| Walters (2021) | To investigate the relationship between physical, sexual and emotional violence and controlling behaviors on early initiation of breastfeeding and continued breastfeeding. | Malawi, Tanzania, Zambia, AFR | Cross sectional | Yes–DHS |
| Shamu (2018) | To analyze the relationship between IPV at different points in the mother's life and maternal and child health effects, including LBW. | Zimbabwe, AFR | Cross sectional | No |
| Laelago (2017) | To investigate the relationship between IPV during pregnancy and adverse birth outcomes in a region of Ethiopia. | Ethiopia, AFR | Cross sectional | No |
| Taft (2015) | To investigate how violence impacts women's reproductive health and infant/ child mortality and health in Timor-Leste. | Timor Leste, SEAR | Cross sectional | Yes–DHS |
| Jaraba (2019) | To investigate the relationship between violence during pregnancy and adverse birth outcomes such as LBW and preterm birth. | Colombia, AMR | Cross sectional | Yes–DHS |
| Assefa (2012) | To identify factors which contribute to LBW in rural Ethiopian communities. | Ethiopia, AFR | Observational cohort study | No |
| Alemu (2019) | To determine factors impacting LBW in the Kambata-Tembaro region of Ethiopia. | Ethiopia, AFR | Cross sectional | No |
| Sobkoviak (2012) | To examine the relationship between maternal exposure to DV and anthropometric data in children under five in Liberia. | Liberia, AFR | Cross sectional | Yes–DHS |
| Berhanie (2019) | To investigate the relationship between IPV and adverse neonatal outcomes in Ethiopia. | Ethiopia, AFR | Case Control | No |
| Rico (2011) | To examine the relationship between IPV among mothers and child stunting and mortality in five LMICs. | Egypt, Honduras, Kenya, Malawi, Rwanda, EMR, AMR, AFR | Cross sectional | Yes–DHS |
| Chai (2016) | To assess the relationship between maternal IPV and their children's nutritional outcomes from 29 different countries. | Azerbaijan, Bangladesh, Bolivia, Burkina Faso, Cambodia, Cameroon, Colombia, Dominican Republic, Gabon, Ghana, Haiti, Honduras, India, Kenya, Liberia, Malawi, Mali, Mozambique, Nepal, Nigeria, Peru, Republic of Moldova, Rwanda, Sao Tome and Principe, Timor-Leste, Uganda, United Republic of Tanzania, Zambia, Zimbabwe, AFR, AMR, SEAR | Cross sectional | Yes–DHS |
| Abujilban (2017) | To investigate how IPV during pregnancy impacts the risk of negative birth outcomes among women in Jordan. | Jordan, EMR | Case Control | No |
| Khan (2021) | To identify the socio-demographic risk factors correlated with severe child malnutrition among children younger than five in Bangladesh. | Bangladesh, SEAR | Cross sectional | Yes–DHS |
| Eno (2014) | To explore how DV impacts pregnancy outcomes. | Nigeria, AFR | Prospective case-control | No |
| Hampanda (2016) | To investigate how IPV against HIV-positive women impacts infant feeding practices | Zambia, AFR | Cross sectional | No |
| Islam (2017) | To investigate the relationship between psychosocial factors, such as IPV, on EBF among Bangladeshi mothers. | Bangladesh, SEAR | Cross sectional | No |
| Pun (2019) | To assess the relationship between DV, LBW and preterm birth among women in Nepal. | Nepal, SEAR | Cross sectional | No |

(*Continued*)

**Table 2.** (Continued)

| Article | Research Question | Country, Region | Study design | Used secondary data Yes/ no If yes, which dataset? |
|---|---|---|---|---|
| Das (2020) | To investigate the determinants of stunting among children younger than 2 years. | India, SEAR | Cross sectional | Yes–Trial to evaluate impact of community resource centres on maternal and child health and nutrition |
| Mezmur (2021) | To examine adverse fetal outcomes (including LBW) among pregnant teenagers and women in rural East Africa. | Ethiopia (Eastern), AFR | Cross sectional | No |
| Arcos (2001) | To document the impact of prior and concurrent maternal exposure to DV on intrauterine growth, birth outcomes in a cohort of women followed longitudinally from pregnancy through delivery. | Chile, AMR | Prospective cohort | No |
| Arcos (2003) | To document the impact of pre and intra pregnancy exposure to DV on infant growth outcomes at age 11 months in children born to women who were followed longitudinally from pregnancy through delivery. | Chile, AMR | Prospective cohort | No |
| Ruiz Grosso (2014) | To assess the association between IPV and chronic malnutrition in children under five years old. | Peru, AMR | Cross sectional | Yes–DHS |
| Faramarzi (2005) | To assess the incidence of self-reported physical, emotional and sexual violence in pregnancy and describe the association with maternal complication and birth outcomes. | Iran, EMR | Cross sectional | No |
| Ferraro (2017) | To determine if there is a measurable association between combined psychosocial factors, specifically DV and mental disorders, and birth outcomes, specifically birth nutritional status and preterm delivery. | Brazil, AMR | Prospective cohort | No |
| Hasselmann (2016) | To investigate the role of IPV in the early interruption of EBF in the first three months of life | Brazil, AMR | Prospective cohort | No |
| Dolatian (2016) | To determine the economic and psychological determinants of birthweight. | Iran, EMR | Prospective cohort | No |
| Lobato (2018) | To evaluate whether psychological IPV during pregnancy is a risk factor for intrauterine growth restriction (IUGR). | Brazil, AMR | Cross sectional | No |
| Mahmoodi (2019) | To determine the factors that predict LBW. | Iran, EMR | Prospective cohort | No |
| Nojomi (2006) | To determine the prevalence of physical abuse in pregnant women and to assess association between physical violence during pregnancy and maternal complications and birth outcomes. | Iran, EMR | Cross sectional | No |
| Hasselmann (2006) | To explore the role of IPV among caregivers as an independent risk factor for severe and acute malnutrition. | Brazil, AMR | Case Control | No |
| Mezzavilla (2016) | To investigate the association between physical IPV and LBW. | Brazil, AMR | Cross sectional | No |
| Nejatizade (2017) | To assess the prevalence of DV in pregnant women and maternal and infants' outcomes. | Iran, EMR | Cross sectional | No |

(Continued)

**Table 2.** (Continued)

| Article | Research Question | Country, Region | Study design | Used secondary data Yes/ no If yes, which dataset? |
|---|---|---|---|---|
| Nunes (2011) | To estimate the prevalence of psychological, physical and sexual violence during current pregnancy and at any time in life; to identify demographic, obstetric, behavioral characteristics and depressive symptoms of those at most risk; and, to assess the impact of violence during pregnancy on newborn outcomes. | Brazil, AMR | Prospective cohort | No |
| Nasreen (2019) | To investigate the independent effect of maternal Antepartum depressive and anxiety symptoms on LBW, preterm birth and CS or instrumental delivery among women in east and west coasts of Malaysia. | Malaysia, WPR | Prospective cohort | No |
| Moraes (2011) | To investigate the role of severe physical violence during pregnancy between intimate partners in early cessation of exclusive breast-feeding. | Brazil, AMR | Cross sectional | No |
| Ribeiro (2021) | To verify whether recurrent violence, violence with pregnancy complications, and IPV against pregnant women are associated with shorter exclusive breastfeeding up to the infant's 6[th] month and breastfeeding up to the 12[th] month of life. | Brazil, AMR | Prospective cohort | Yes–Brazilian Ribeirão Preto and São Luís Birth Cohort Studies (BRISA) |
| Caprara (2020) | To examine the influence of DV against pregnant women on early complementary feeding and associated factors. | Brazil, AMR | Longitudinal | No |
| Abdollahi (2015) | To determine the prevalence of physical violence against women by an intimate partner during pregnancy, and to assess the impact of this physical violence on pregnancy outcomes. | Iran, EMR | Prospective cohort | No |
| Khodakarami (2009) | To assess the pregnancy outcomes of abused vs non-abused women. | Iran, EMR | Cross sectional | No |
| Abadi (2013) | To investigate associations of birth weight with sociodemographic variables, domestic violence, ways of coping, social support, and general mental health of Iranian mothers. | Iran, EMR | Cross sectional | No |
| Aristizabal 2022 | To explore the relationship between IPV and breastfeeding practice in Colombia | Colombia, AMR | Cross sectional | Yes-DHS |
| Barnett 2022 | To explore the relationship between IPV and IPV subtypes and growth. Exploring mediators in the IPV growth relationship. | South Africa, AFR | Longitudinal-prospective cohort | Yes- South Africa birth cohort on child health study |
| Chandra 2021 | To explore the relationship between antenatal anxiety, depression, and IPV and birth weight. | India, SEAR | Longitudinal-prospective cohort | Yes- India Maternal Mental Health Study established at the National Institute of Mental Health and Neurosciences, in Bangalore India |
| Avci 2022 | To explore the relationship between domestic violence during pregnancy and cortisol hormone, preterm birth, low birth weight and breastfeeding. | Turkey, EUR | Cross sectional | No |
| Chowdhury 2021 | To observe the risk factors for malnutrition in children under 5 and the socio-demographic determinants. | Bangladesh, SEAR | Cross sectional | Yes-DHS |

(*Continued*)

**Table 2.** (Continued)

| Article | Research Question | Country, Region | Study design | Used secondary data Yes/ no If yes, which dataset? |
|---|---|---|---|---|
| Debele 2022 | To determine the relationship between LBW and physically demanding work during pregnancy along with others factors such as IPV and household food insecurity. | Ethiopia, AFR | Cross sectional | No |
| Doke 2021 | To explore the relationship between adverse pregnancy outcomes and risk factors including domestic violence and others. To assess the rates of adverse pregnancy outcomes and compare tribal vs not tribal and cases vs controls. | India, SEAR | Cross sectional | No |
| Fonseka 2022 | To explore the relationship between child marriage and maternal IPV exposure to stunting in children under 5. Also explored the role as moderator distance to conflict plays in the relationship between maternal child marriage and stunting, and maternal IPV and stunting separately. | Sri Lanka, SEAR | Cross sectional | Yes-DHS |
| Issah 2022 | To explore the relationship between IPV and women and child nutrition in Nigeria. | Nigeria, AFR | Cross sectional | Yes- DHS |
| Woldetensay 2021 | To explore the relationship between depressive symptoms and IPV and social support and infant feeding practices. | Ethiopia, AFR | Longitudinal, prospective cohort | No |
| Okunola 2021 | To explore the relationship between IPV during pregnancy and adverse birth outcomes in south western Nigeria. | Nigeria, AFR | Prospective cohort | No |
| Tesfa 2021 | To explore the prevalence of fetal malnutrition in Ethiopia and factors related to fetal malnutrition. | Ethiopia, AFR | Cross sectional | No |
| UysalYalcin 2022 | To explore the relationship between intimate partner violence and child growth for children under 5. | Turkey, EUR | Cross sectional | No |
| Vachhani 2022 | To explore the health impact on women's and children's health for those women who experience domestic violence and to look at health seeking behavior of those women. | India, SEAR | Cross sectional | No |

AFR, African Region; AMR, Region of the Americas; SEAR, South-East Asian Region; EUR, European Region; EMR, Eastern Mediterranean Region; WPR, Western Pacific Region

Thirteen studies examined the association between any type of IPV experience in the past year and LBW, of which one found a significant association of increased risk for LBW among children born to mothers reporting physical IPV [46]. No other associations were found for LBW and other types of IPV individually or combined for experiencing IPV in the past year.

Twenty-five studies examined the association between experience of any type of IPV during pregnancy and LBW. An overwhelming majority of studies (10 of 13) that examined experience of physical IPV during pregnancy and LBW, found a significantly increased risk of LBW [6, 47–58]. Similarly, ten of thirteen studies examining experience of combined IPV during pregnancy and LBW found that gestational experience of combined IPV significantly increased risk of LBW [6, 47–49, 59–67]. Experience of sexual IPV during pregnancy was significantly associated with increased risk of LBW in three of ten studies that examined this association [6, 47–50, 52, 53, 56, 57, 68]. Four out of the nine studies that examined experience of

**Table 3. Quality assessment.**

| Quality assessment item | Possible responses | n = 86 | Percent |
|---|---|---|---|
| **Introduction** | | | |
| Was the research question or objective in this paper clearly stated? | Yes | 85 | 99% |
| **Methods** | | | |
| Representativeness of the sample | Truly representative of the average in the target population (all subjects or random sampling) | 34 | 39.5% |
| | Somewhat representative of the average in the target population. (non-random sampling) | 4 | 4.70% |
| | Selected group (i.e. clinic-based sampling in hospital) | 46 | 53.5% |
| Was the target/reference population clearly defined? | Yes | 77 | 89.5% |
| Is the sampling strategy relevant to address the research question? | Yes | 70 | 81.4% |
| Non-Respondents | (1) Comparability between respondents and non-respondents' characteristics is established and the response rate is satisfactory | 35 | 40.7% |
| | (2) The response rate is unsatisfactory or the comparability between respondents and non-respondents is unsatisfactory | 8 | 9.30% |
| | (3) No description of the response rate or the characteristics of the responders and the non-responders. | 43 | 50.0% |
| Is the violence exposure measured rigorously? | Yes | 79 | 91.9% |
| **Results** | | | |
| Is the nutrition outcome measured rigorously? (Internal validity) | Yes | 75 | 87.2% |
| Were key potential confounding variables measured and adjusted statistically for their impact on the relationship between exposure(s) and outcome(s)? | Yes | 73 | 84.9% |
| Statistical test | The statistical test used to analyze the data is clearly described and appropriate and the measurement of the association is presented including confidence intervals and the probability level (p value) | 83 | 96.5% |
| | The statistical test is not appropriate not described or incomplete. | 3 | 3.50% |

emotional or psychological IPV or controlling behaviors found significantly increased risk of LBW [6, 47–50, 52, 55, 56, 68].

## Associations with breastfeeding practices

Fig 2B shows the associations found between experience of any type of IPV, in a lifetime, in the past year, or during pregnancy and feeding practices, with n = 18 focused on breastfeeding, one focused on minimum acceptable diet and one on infant feeding practices which included breastfeeding 0 to 6 months and feeding practices 6 to 12 months measured every three months.

Seven studies examined lifetime experience of physical IPV and suboptimal breastfeeding practices, of which five found significant positive associations. Two, which were multi-country studies, found positive associations in some countries and not others [5, 69–74]. Of eight studies examining combined IPV and suboptimal breastfeeding practices, significant associations were reported in five studies [69, 70, 73–75]. Lifetime experience of sexual IPV and measures for suboptimal breastfeeding practices were examined in six studies, of which five found significant positive associations [5, 69–71, 73, 74]. In six of 14 associations explored, a positive association was found between suboptimal breastfeeding practices and experience of emotional/ psychological/ verbal IPV or controlling behavior. Associations varied depending on country where the study took place [5, 69–74, 76].

Six studies examined the association between experience of any type of IPV in the past year and suboptimal breastfeeding practices. One out of the two that assessed physical IPV in the past year found significant association with suboptimal breastfeeding [77, 78]. Sexual IPV was

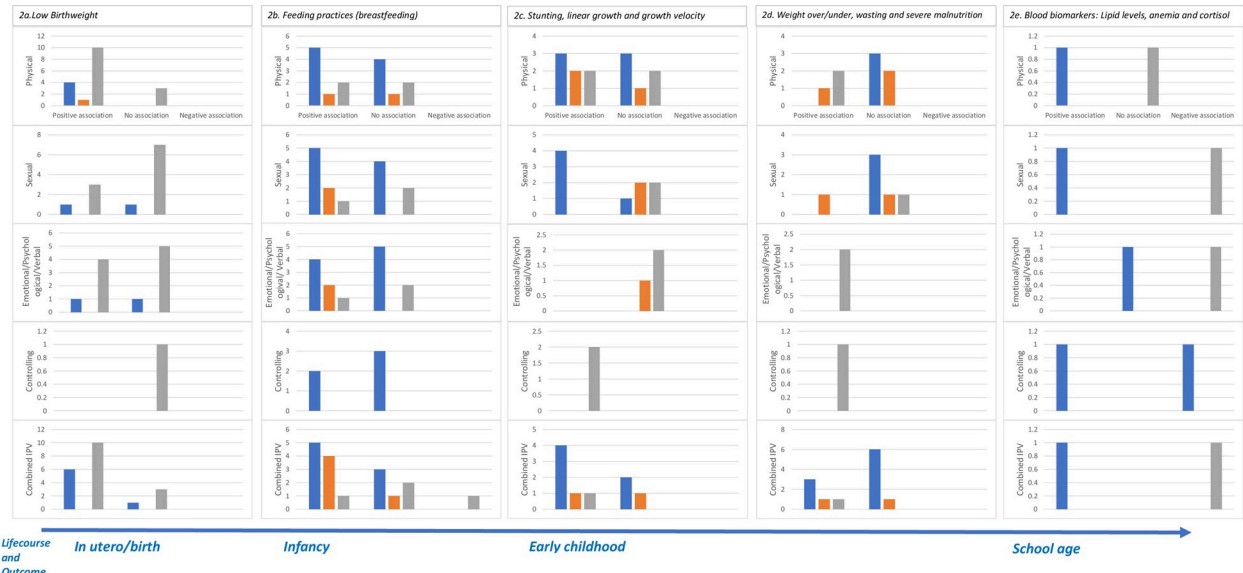

**Fig 2. Harvest plot of maternal experience of IPV and child nutrition and growth outcomes.** Harvest plot depiction of associations between maternal exposure to IPV and child outcomes related to growth and nutritional status, by timepoint of exposure. The figure below the plot shows a life-course and outcome timeline. (a) Shows associations with low birthweight. (b) Shows associations with feeding practices including breastfeeding. (c) Shows associations with stunting, linear growth and growth velocity. (d) Shows associations with underweight/ wasting/ severe malnutrition. (e) Shows associations with lipid biomarkers. The blue bar in the graphs depict maternal lifetime exposure to IPV, the orange bar depicts exposure to IPV in the past year and the gray bar depicts exposure to IPV during pregnancy.

significantly associated in the two studies to examine the association [77, 78]. Of the three studies examining experience of combined IPV in the past year and suboptimal breastfeeding, three studies reported a positive association, where one study looked at pre-lateral and delayed initiation of breastfeeding separately and found a negative and positive association respectively [77–79]. Both studies examining emotional/ psychological/ verbal/ and controlling behavior with sub-optimal breastfeeding, had a positive association [77, 78].

Two of the four studies that examined the impact of experiencing physical IPV during pregnancy on breastfeeding practices, showed that delayed initiation of breastfeeding and lack of exclusive breastfeeding were significantly associated with experience of physical IPV [72, 77, 80, 81]. Three studies examined associations between experiencing combined IPV during pregnancy and suboptimal breastfeeding practices. Two showed no association and one reported a statistically significant negative z score, where women who experienced combined IPV during pregnancy had less confidence in their breastfeeding practices [65, 72, 77], while breastfeeding and experience of sexual IPV during pregnancy showed positive significant associations in one of three studies that examined that association [72, 77, 80]. An association between experience of emotional/ psychological/ verbal IPV and suboptimal breastfeeding practices was found in one of three studies [72, 77, 80] with none looking at controlling behavior during pregnancy and breastfeeding.

## Associations with feeding practices in infancy and second year of life

Two studies examined feeding practices beyond breastfeeding, where one analyzed data collected through the Ethiopian 2016 DHS, examining the association of maternal caregivers' experience of combined IPV (including physical, sexual and emotional violence) on minimum acceptable diet (a composite of the UNICEF indicators for minimal dietary diversity and minimal meal frequency) in children ages 6–23 months and found a significant association between

maternal IPV and poorer minimum acceptable diet [82]. The other study, also based in Ethiopia, but using data from the Improve Nutrition and Economic (ENGINE) opportunities birth cohort study, found the maternal experience of combined IPV increased the risk to poorer infant feeding practices [83].

## Associations with indicators of undernutrition

**Stunting, linear growth and growth velocity.** Fig 2C shows the associations between a maternal caregiver's experience of any type of IPV and stunting, severe stunting, decreased linear child growth, or growth velocity in children, as examined in fifteen studies. Overall, maternal experience of any type of IPV was significantly associated with a child's stunting, with the strongest findings among those studies examining maternal lifetime experience of any type of IPV, rather than to a mother's experiencing any type of IPV (only) during the last year as might be expected given the chronicity of the malnutrition represented by stunting.

Maternal lifetime experience of physical IPV was associated with a child's stunting or decreased growth velocity in three out of the five studies that examined this association [42, 84–87] and was associated with severe stunting in the one study that explored this association, although there was no association in the same study but for other locations [85]. In the six studies that examined lifetime experience of combined IPV and stunting during childhood, four found that risk of stunting was significantly elevated among children whose maternal caregiver reported a lifetime experience of combined IPV [42, 84–88]. Similarly, in the two studies that examined maternal lifetime experience of combined IPV and a child's risk of severe stunting, one found significantly increased risk of severe stunting [38, 85], and the same experience significantly predicted decreased growth velocity (height) in the only study in which it was examined [44]. Maternal caregivers' lifetime experience of sexual IPV was associated with a child's stunting in four of five studies [84–87, 89]. Lifetime experience of emotional/ psychological/ verbal or controlling behavior was not analyzed with any of the stunting or linear growth outcomes.

Three studies examined maternal caregivers' experience of physical IPV in the past year and stunting, of which two demonstrated a significant association with stunting [90–92]. Of two studies to analyze an association between experience of combined IPV in the past year with stunting, one reported a significant association [89, 91]. Two studies looked at an association with experience of sexual IPV in the past year but no significant association was found [91, 92]. One study explored the association between stunting and maternal past year experience of emotional/ psychological/ verbal or controlling behavior and found no significant association [92].

Two studies examined the association between experience of any type of IPV during pregnancy, and linear child growth using height for age z scores. An association was found for physical IPV in one study [93]. There was no association found for experience of sexual, or emotional/psychological/verbal IPV, however, both controlling behaviors and combined IPV were significantly associated with a child's decreased height for age in two studies [93, 94].

**Underweight/wasting/severe malnutrition.** Fig 2D shows the associations between maternal caregivers' experience of any type of IPV and child growth indicators such as underweight, growth velocity, wasting, severe malnutrition, low BMI, fetal malnutrition, overweight and the composite index of anthropometric failure [CIAF] that were analyzed in fifteen studies.

Several studies of lifetime experience of any type of IPV and growth indicators showed no significant associations: physical IPV and underweight [42, 84, 86], and lifetime experience of sexual IPV and underweight [89]. Lifetime experience of combined IPV and underweight

showed one positive significant association out of two studies that analyzed this relationship [84, 88]. The only study to look at experience of combined lifetime IPV and growth velocity in weight found a significant association [44]. The two studies that looked at associations between physical IPV and wasting found no significant associations [42, 86]. Three studies found non-significant associations between lifetime experience of combined IPV and wasting [42, 86, 88]. A significant association was found in the one study to look at combined IPV and CIAF [95]. The two studies to analyze lifetime combined IPV and severe malnutrition found no significant association [38], and there were no associations between lifetime experience of sexual IPV and wasting in the two studies exploring this association [84, 86].

Two studies examined maternal caregivers' experience to physical IPV in the past year and a child being underweight; neither found a significant association [90, 91]. Of the two studies that explored associations between past year physical IPV and wasting, one of them found a significant association [90, 91]. A significant association was found in the only study to analyze a relationship between past year experience of physical IPV and low BMI [90]. Experience of combined IPV in the past year and being underweight yielded a significant association in one of two studies that assessed this association [89, 91]. One of two studies looking at past year experience of combined IPV and wasting found a significant association [89, 91]. Maternal caregivers' experience of sexual IPV in the past year and a child being underweight was significant in one of two studies to examine this association [89, 91], and showed no association with wasting in the two studies to explore this relationship [89, 91]. No studies looked at an association between underweight/ wasting outcomes with maternal past year experience of emotional/ psychological/ verbal or controlling behavior.

The only study to explore a relationship between maternal caregivers' experience to physical IPV during pregnancy and a child being underweight found a significant association [68]. A significant association was found between experience of physical IPV during pregnancy and wasting in the single study to explore this relationship [93]. Experience of combined IPV during pregnancy and fetal malnutrition yielded a significant association in the one study that analyzed this relationship [96]. Maternal experience of sexual IPV during pregnancy and a child being underweight was not found to be significant in the one study to examine this association [68]. The one study to examine maternal experience of emotional/psychological and verbal IPV during pregnancy and underweight found a significant association [68], the relationship with the same type of IPV and wasting was also found to be significant for the one study to analyze it [93]. One study analyzed the relationship between overweight and maternal experience of emotional/psychological and verbal IPV and controlling behavior separately and found significant associations for both relationships [93].

## Associations with blood-based biomarkers

Four studies included measures of blood-based markers of child nutrient intake. Fig 2E shows the associations between maternal experience of any type of IPV and anemia, or lipid profiles in children. One large longitudinal study by Ziaei et al., 2019, in Bangladesh found that children's lipid profiles collected at age 10 years were significantly worsened among children whose mothers reported lifetime experiences of physical, sexual, controlling behaviors or combined forms of IPV [97]. The association of child lipid profiles with maternal IPV differed depending on the type of IPV as well as by whether the maternal experience was prenatal, or occurred postnatally during 10 years after the child's birth [97]. Some lipids appeared more sensitive to specific forms of IPV. For example, triglyceride levels increased with physical but no other forms of IPV, regardless whether exposure was pre- or postnatal. ApoA levels worsened (decreased) with prenatal maternal exposure to combined IPV, isolated sexual IPV, and

controlling behaviors and LDL/HDL ratios worsened (increased) among children whose mothers were exposed to sexual, but not other forms of IPV [97]. Controlling behaviors were associated with worsened lipid profiles if the mother experienced them prenatally, but surprisingly, when mothers' exposure to controlling behaviors was postnatal, children had *improved* lipid profiles (lower LDL, LDL/HDL and cholesterol) at age 10, which was hypothesized to result from the paradoxical effect of mothers' increased time at home with their children [97].

In the two studies that examined a child's anemia as an outcome, none found an association with maternal caregivers' experience of IPV. However, the studies that examined this association only examined the maternal experience of IPV during the past year, or lifetime, rather than examining the maternal exposure during a time period that would correspond more directly to the temporality of a single measure of hemoglobin. Additionally, the studies did not examine other markers that would permit differentiation of iron deficiency from other types of anemia. Examining experience of physical IPV in the past year and child's anemia showed no association in the only study to look at the relationship [90]. Lifetime experience of combined IPV was not associated with childhood anemia in children 6 to 59 months in the only study that assessed this relationship [42]. No other associations of anemia were reported with any other type of maternal IPV or combined IPV experience in the past year. No other associations were reported between maternal caregivers' experience of IPV during pregnancy and child anemia or lipid biomarker outcomes.

One study analyzed the relationship between caregivers' experience of physical IPV during pregnancy and increased cortisol levels in the child and did not find a significant association. However, this same study looked at the associations between a caregivers' experience of combined IPV, sexual IPV and emotional/psychological and verbal IPV and found cortisol levels to be significantly higher for babies where mothers had experienced these types of violence during pregnancy [65].

## Discussion

This REA represents an important update to prior reviews focusing on the association between a maternal caregivers' experience of any type of IPV and children's nutrition outcomes. While the global evidence-base remains biased towards evidence gathered in high-income contexts, our review indicates that the past decades of research and analysis has generated significant insights into the associations between IPV and various nutrition outcomes in LMICs in all regions.

In examining potential mechanisms for how IPV can contribute to each of the groups of outcomes, LBW, feeding practices and various indicators of child growth and malnutrition, there is scope for hypothesizing causal pathways between experience of IPV by a maternal caregiver and nutritional/growth outcomes. Based on the data gathered from multiple studies for this review, there is a suggestion of a linear impact of experience of IPV during pregnancy on physiological pathways affecting fetal growth. This is supported by the strong and consistent associations between mothers' experience of physical or combined IPV and LBW, while the impact of emotional/psychological IPV or sexual IPV may work through an intermediary in the pathway to LBW.

Although thirteen studies assessed the association between maternal experience of any type of IPV and LBW in the past year, only one study found a positive association with physical IPV [46]. In contrast, positive associations were found in six of seven studies examining lifetime combined IPV and LBW [39–45]. Similarly, 19 of 25 studies found a positive association between maternal experience of any type of IPV during pregnancy and LBW. The notable difference between experience of any type of IPV during a lifetime and during pregnancy vs experience of any type of IPV in the past year indicate the importance of taking into account

timing of when a maternal caregiver experiences IPV to understand causal pathways to child nutrition outcomes.

Our findings find a significant association between mother's experience of IPV and breast-feeding. The evidence-base confirms the hypothesis that women who have experienced IPV face psychological or physical barriers to breastfeeding, known as the deficit hypothesis, as opposed to the compensatory hypothesis, whereby women who have experienced IPV may be more sensitive or responsive to their children's needs and therefore have higher odds of early initiation of breastfeeding, exclusive breastfeeding and/ or continued breastfeeding than women not exposed to IPV [75]. Unfortunately, most studies did not specifically focus on IPV occurring between birth and the interview. This is an important period to examine in new research, given that the peak risk periods during which experience of IPV experience can impact adherence to recommended practices in breastfeeding could be immediately following delivery as well as during pregnancy. Breastfeeding is critical for child health and growth, and contributes improved long-term health, including cognitive capacity and survival [98–100]. Among its multiple benefits are protection from infections during infancy, as well as providing hygienic food in settings with limited potable water (especially relevant in humanitarian settings), in fragile settings, the life-saving protection of breastfeeding is thus especially critical. Conflicts, natural disasters and epidemics contribute to forced migration resulting in heightened food insecurity, limited access to clean water, and disruptions to basic services, leaving women and children especially vulnerable. In these settings, breastfeeding guarantees a safe, nutritious and accessible food source for infants and a protective shield against infectious disease and death [100]. Preventing, managing or mitigating IPV can improve breastfeeding practices. Initiation of breastfeeding promotion interventions that reach mothers even before conception and recognizing the impact of a maternal experience of IPV on her breastfeeding practices, can impact child health indicators throughout childhood. Although some literature shows that breastfeeding is affected by sex feeding preference, where the mother may breast-feed a male more than a female child [101, 102], no articles on feeding preference and IPV were identified that met the inclusion criteria for this review. In addition, we found a surprising lack of sex disaggregation of child nutrition outcomes, which prevents proper analysis on sex based preferential breastfeeding and other feeding practices.

Findings indicate that experience of IPV by a maternal caregiver is associated with impaired child growth and indicators of acute and chronic malnutrition. Despite some lack of comparability of associations, the studies included in this review suggest that maternal experience of IPV affects several growth indicators in preschool children, however, the literature is extremely limited in its examination of time intervals that can fully inform our understanding of any potential underlying mechanism. Despite this, the literature that includes stunting as an outcome does suggest that maternal caregivers' lifetime experience of IPV is more predictive of stunting than past year experience of IPV. This is suggestive of a pathway consistent with evidence where lifetime IPV can lead to inadequate nutrition and reduced care practice. The literature reviewed suggested that maternal caregivers' experience of physical alone, sexual alone or combined IPV posed a greater risk for a child's stunting, rather than other types of IPV. For more acute growth indicators, other types of IPV (physical) and a shorter period (past year) appear to pose greater risk. To better examine the pathways through which stunting and underweight may result from maternal caregivers' experience of IPV, further work should examine underlying potential mechanisms, including examination of feeding patterns. The finding of stunting and lifetime maternal experience of IPV, may be particularly amenable to earlier intervention and strategies for prevention. In particular, capturing reporting of lifetime experience shows an underlying chronicity of exposure and long developmental window for stunting, which may have a physiologically different impact.

Though few studies examined maternal caregivers' experience of IPV alongside children's nutrient biomarkers in blood, the results from the one study suggests a potential causal pathway for lipid dysregulation (at age 10) leading to metabolic syndrome and contributing to later obesity resulting from maternal caregivers' experience of physical and sexual IPV [97]. One other study found an increase in cortisol levels for newborns where the maternal caregiver had experienced several types of IPV during pregnancy [65] and in another study, a higher cortisol level in the mother, as well as the baby, was associated with lower birth weight [103]. Notably absent from the literature were studies examining the impact of maternal caregiver IPV on a child's anemia during early childhood when children are reliant on caregiver feeding. Similarly, examination of maternal caregiver exposure to IPV and child measures of lipid dysregulation during puberty and adolescence may further elucidate potential pathways contributing to a child's less healthy food choices later in life.

Results of the REA indicate the importance of context in associations between maternal caregiver experiencing IPV and child nutrition outcomes. In some multi-country studies, associations were found in one context and not another. For example, Misch et al., reported significant associations between physical IPV and delayed initiation of breastfeeding in Tanzania, Malawi and Zimbabwe and no association in Ghana, Liberia, Nigeria, Zambia and Kenya. Further supporting this example they found an association between sexual IPV experience and delayed initiation in Zambia but not in other countries [71]. Despite consistent use of methodology, IPV recall period and measurement of both IPV and nutrition outcomes in this multi-country study, the variation in results between contexts indicates that important, and likely unmeasured, contextual factors influence the association between indirect IPV exposure and child nutrition outcomes. Variance in associations occurred in a number of multi-country studies. In addition, intra-country variation in single-country studies may be substantially masked by country-level analyses of DHS data, in particular.

The REA also identified significant but addressable knowledge gaps, one, in the examination of risk during time periods preceding occurrence of the outcomes, and secondly in considering time periods that would be physiologically and developmentally meaningful for the progression of the underlying processes measured in the outcomes chosen. For example, although a large number of studies examined risk associated with experience of IPV during pregnancy, no studies considered the relevance of timing during pregnancy considering fetal development when examining the association between experience of IPV against a maternal caregiver and a child's LBW. Additionally, studies examining the impact of physical IPV did not differentiate physical IPV involving direct abdominal trauma. Similarly, in those studies examining experience of IPV and breastfeeding, there was no documentation of IPV occurring during the child's infancy (i.e. post-natal). Studies examining stunting or growth velocity similarly did not disaggregate maternal caregivers' experience of IPV after the child was born. Many of these limitations may be due to the reliance on secondary data from large standardized regional/ national surveys. Studies reporting primary data collection though smaller were generally more able to provide insight into underlying processes.

Other gaps in the evidence-base identified in the REA include that only one study considered impact on young children by considering feeding practices in children above 6 months old, and using the UNICEF indicators for minimal acceptable diet (a composite of minimal dietary diversity, and age specific minimal meal frequency) [82]. These data are available in DHS datasets, and there is potential to conduct several secondary analyses of DHS data, including cross-country comparisons, to illustrate the associations between IPV and minimal acceptable diet. Studies in infants, children, and adolescents should include measures of diet diversity and diet adequacy as this is understudied (or completely unstudied). Given our findings regarding associations of IPV with breastfeeding, as well as the evidence from the two studies that examined

minimal acceptable diet, it is highly plausible that feeding/ patterns of dietary intake in children older than 6 months will also be affected by either direct or indirect exposure to IPV. Strategies to prevent or remediate poor growth in infancy could be directly informed by including measures of the impact of IPV on dietary adequacy in infants and children.

A notable limitation of the evidence-base is the reliance on cross-sectional study designs as they do not allow for follow up to other outcomes as the child grows. The measurement and categorization of confounders, and inclusion of confounders, varied widely between studies, limiting direct comparison of results even if studies were using the same exposure and outcome measures. While measurement of IPV across the evidence-base was generally of high quality, inconsistency in grouping of types of IPV and different use of recall periods (lifetime, past year and during pregnancy) across studies led to lack of comparability of many associations identified in the studies. The secondary data analyses have inherent limitations because the studies are not designed to examine the association of specific types of IPV against a maternal caregiver with specific child outcomes. Many of the datasets could be used to examine these potential pathways with more precision by restricting the analyses to those in whom the time period of measured exposure corresponds to, a time period that is biologically and plausibly relevant for the outcome of interest.

There are various programmatic implications resulting from the REA. The evidence highlights the potential for greater synergies across GBV, nutrition, and maternal and child health programs. By identifying entry points within nutrition programs to provide support to survivors and refer them to specialized GBV services, such programs could improve well-being for both women and children. Equally, there are also opportunities to integrate more information on nutrition into GBV services to ensure that women who experience GBV are able to access nutrition services when their children are at risk. The evidence provides further support for advancing GBV prevention, given the potential lifelong impacts of IPV on child nutrition, and therefore, health and wellbeing. Effective approaches, such as UNICEF's Communities Care model, hold promise to reduce GBV and thereby also contribute to better nutrition outcomes [104]. The findings of the assessment illustrate that incorporating GBV considerations into nutrition policies and programming is relevant to the protection and promotion of child health. Taking action to proactively identify and mitigate GBV-related risks helps make nutrition programming more effective especially in humanitarian contexts where the risk of GBV is higher.

## Limitations

The results of the REA should be interpreted in light of some limitations, including that we conducted single screening of titles and abstracts (a single reviewer determined whether to include or exclude), and that data extraction was conducted by a single reviewer for each article. This methodology is in line with recent guidance regarding how to adapt systematic review methodology to the purposes of a rapid evidence assessment [105]. A second team member checked the data extraction for completeness and accuracy, and title/ abstract screening was conducted after 10% of the sample was double-screened, to ensure inter-rater reliability and consistency in application of inclusion and exclusion criteria. Studies in languages other than English and Spanish were excluded from the review due to language limitations of the team. The most significant limitation in interpretation of the results is that we did not weight studies according to quality or sample size. This level of quality assessment was beyond the scope of the REA in terms of timeline and resources. The findings regarding type and direction of associations, and patterns of associations between specific types of IPV and specific nutrition outcomes, should therefore be interpreted with this limitation in mind. Despite these limitations, this review represents an important update to prior evidence synthesis of the associations

between maternal IPV and child growth and nutrition. By capturing the past few years of research, which has seen an enormous growth in published analyses, represents a comprehensive and up-to-date synthesis of this literature.

## Conclusion

Our review identified maternal caregivers' experience of IPV to be directly associated with low birthweight and breastfeeding practices, and also significantly associated with impaired fetal, infant, and child growth and indicators of acute and chronic malnutrition. There are opportunities that could be more strategically leveraged to better understand how experience of IPV can impact child growth indicators. Future studies could be designed to specifically evaluate experience of IPV during periods of risk relevant to specific growth outcomes, and to disaggregate analysis of outcomes by child sex. Despite some lack of comparability of associations identified in the studies, and differing patterns between specific types of IPV and different nutrition outcomes, there is ample evidence that IPV negatively affects a broad range of child nutrition outcomes. Acknowledging IPV as a risk factor within nutrition and child health programs holds the potential to significantly improve the nutrition, health and well-being of both children and women. Linking maternal caregivers' experience of IPV to child malnutrition and growth pathways creates a space for conversations around integrated program models and the specific need for more development, testing and refinement of these models in LMIC settings. Approaches including targeted IPV screening of women whose children are undergoing treatment for malnutrition through supplementary feeding programs, training pre-natal health care workers and lactation specialists to identify signs of IPV amongst women, and strengthening referral networks between health and nutrition services and IPV response services are all key actions that can be explored in LMICs.

## Supporting information

**S1 Appendix. Search domains and terms for all databases.**
(DOCX)

**S1 Table. Study results.**
(DOCX)

**S2 Table. Quality assessment.**
(DOCX)

## Acknowledgments

We thank Lily Coll for her support in data extraction.

## Author Contributions

**Conceptualization:** Silvia Bhatt Carreno, Manuela Orjuela-Grimm, Luissa Vahedi, Elisabeth Roesch, Christine Heckman, Andrew Beckingham, Megan Gayford, Sarah R. Meyer.

**Formal analysis:** Silvia Bhatt Carreno, Manuela Orjuela-Grimm, Sarah R. Meyer.

**Funding acquisition:** Christine Heckman.

**Investigation:** Silvia Bhatt Carreno, Manuela Orjuela-Grimm, Sarah R. Meyer.

**Methodology:** Silvia Bhatt Carreno, Manuela Orjuela-Grimm, Luissa Vahedi, Sarah R. Meyer.

**Project administration:** Elisabeth Roesch.

**Supervision:** Sarah R. Meyer.

**Validation:** Sarah R. Meyer.

**Writing – original draft:** Silvia Bhatt Carreno.

**Writing – review & editing:** Silvia Bhatt Carreno, Manuela Orjuela-Grimm, Luissa Vahedi, Elisabeth Roesch, Christine Heckman, Andrew Beckingham, Megan Gayford, Sarah R. Meyer.

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
