## [Decision Letter · Decision Letter 0]

17 Aug 2023

PONE-D-23-02037Linkages between maternal experience of intimate partner violence and child nutrition outcomes: A rapid evidence assessmentPLOS ONE

Dear Dr. Meyer,

Thank you for submitting your manuscript to PLOS ONE. After careful consideration, we feel that it has merit but does not fully meet PLOS ONE’s publication criteria as it currently stands. Therefore, we invite you to submit a revised version of the manuscript that addresses the points raised during the review process.

Please note that, we have received only one reviewer comments on your manuscript. We invited other reviewer but could not get positive response. Therefore, we are making decision based on reviewer and my comments. Please submit your revised manuscript by Oct 01 2023 11:59PM. If you will need more time than this to complete your revisions, please reply to this message or contact the journal office at plosone@plos.org. Please include the following items when submitting your revised manuscript:A rebuttal letter that responds to each point raised by the academic editor and reviewer(s). You should upload this letter as a separate file labeled 'Response to Reviewers'.A marked-up copy of your manuscript that highlights changes made to the original version. You should upload this as a separate file labeled 'Revised Manuscript with Track Changes'.An unmarked version of your revised paper without tracked changes. You should upload this as a separate file labeled 'Manuscript'.

We look forward to receiving your revised manuscript.

Kind regards,

Pradeep Kumar, Ph.D.

Academic Editor

PLOS ONE

Journal Requirements:

2. Please identify your study as "systematic review" in the title of your manuscript.

Additional Editor Comments:

**Abstract:**

***Line number: 52-58:*** The authors should provide a programmatic implications/policy with the special focus on low and middle-income countries.

**Introduction:**

Overall observation:

Within the children which particular age group author focuses, its is missing in the introduction section.It would be good if author provide few literatures on gender wise differentiation on any child nutrition, IPV.Overall in whole introduction section need to provide country specific example it would and matched with low and middle-income countries.

***Line number: 89-93:*** It’s better to cite recent systematic review study with the special focus on low and middle-income countries. Try to include systematic review which was published after 2020.

***Line number: 94-98:*** Author provides the reference of first three line (line number: 94-96) also need to brief the sentences. No need to elaborate one single statement into different references. 

***Line number: 100-103:*** The statement related to mental health among adolescents are very crude, Author should provide country specific case study or evidence also remove the word ‘Some children’ try to include better and clear words. 

**Methods:**

***Line number: 155-156:*** I would suggest to use 'PICOTS' framework to summaries the eligibility criteria.

***Line number: 157-160:*** It is good to provide the search terms in these three Medline, Embase, and Global Health electronic databases. This can be added as an Appendix A. Provide the search strategy with proper keywords, output and date of search. Overall appendix A need to revised and prepared as per the proper systematic review protocol.

***Line number: 157-160:*
**No need to provide ‘IPV’ definition in this section, either it is better to include in introduction section.

***Line number: 176-179:*** Did the researchers use an application to support and track the screening process of the identified 3037 studies. If yes, please state what was used and how. Who is ‘one reviewer’, ‘two reviewers’, ‘third independent reviewer’? It would be better to provide the short indication name of the author after using these words.

It would be clear, if researcher prepare the conceptual framework and include in the methods section.

**Data extraction and assessment of quality of included studies:**

How was the data extracted and validated between the researchers.

***Line number: 187-190:*
**How researcher was used four quality assessment tools in a single study? How did you validate your quality assessment based on using different questions? How did you calculate quality score? Overall, this section needs to revised and write very carefully and provide full explanation of how researcher uses these four different quality assessment tools? Is there any previous systematic review or rapid review that used same method of quality assessment if yes provide the references?

Author should attach quality assessment of each study and their score in a single file and attach in a supplementary file.Risk of bias also missing in the method as well as result section, try to incorporate, in the revised manuscript.

**Results**

***Line number: 204-205***: S2 Table. Study Results should be categorized into four different categorise, namely:

a) Fetal growth

b) Infant feeding

c) Child growth

d) Nutrient blood markers

Overall, the result section is satisfactory.

**Discussion**

***Line number: 460:*
**Replace the word ‘room’ with other synonym word

***Line number: 469-472:*
**For this particular line *“the studies suggest that maternal experience  of IPV affects several growth indicators in preschool children, however, the literature is extremely limited in its examination of time intervals that can fully inform our understanding of any potential underlying mechanism”* provide the reference.

***Line number: 488-489:*
**For this particular line “In contrast, there were seven out of twelve studies that found a positive association between maternal lifetime experience of any type of IPV and LBW” provide the reference.

***Line number: 495-496:*
**Need to reframe this line “Regarding the association between a mother's experience of IPV and breastfeeding, our findings indicate a significant impact.”

***Line number: 577-578:*
**If this line is indicating the limitation then it should be under the limitation section.

***Line number: 584-585:*
**If this line is indicating the limitation then it should be under the limitation section.

In the discussion section, uniformity is missing, try to accumulate your finding based on your key outcome (Fetal growth, Infant feeding, Child growth, Nutrient blood markers). Also, I observed that between the discussion few line are highlighting the limitation and suggestion try to remove these lines and add those lines in the proper order.Try to add ew lines on strength of this study except only writing the limitation section.

**Conclusion**

The authors should provide a programmatic implications/policy with the special focus on low and middle-income countries. 

Reviewers' comments:

Reviewer's Responses to Questions

**Comments to the Author**

1. Is the manuscript technically sound, and do the data support the conclusions?

Reviewer #1: Yes

2. Has the statistical analysis been performed appropriately and rigorously? 

Reviewer #1: Yes

3. Have the authors made all data underlying the findings in their manuscript fully available?

Reviewer #1: Yes

4. Is the manuscript presented in an intelligible fashion and written in standard English?

Reviewer #1: Yes

5. Review Comments to the Author

Reviewer #1: At the outset, I would like to congratulate the team to work on the relevant topic of IPV of maternal caregivers and child nutrition outcomes which though have been researched several times, needs periodic re-examination. Moreover, the approach to the study i.e., applying REA makes the study unique. It is an extremely well written & well researched article. Although I have a few minor queries, I would be happy to see it published.

1. Define LMIC.

2. Why to include adolescents?

3. Please elaborate line number 159-160

4. Methodology adopted to plot Harvest plot should be explained. Perhaps, after data synthesis (line no. 191- 194), the authors can add a section on the methodology adopted in the article.

6. PLOS authors have the option to publish the peer review history of their article (what does this mean?). If published, this will include your full peer review and any attached files.

Reviewer #1: No

---

## [Author Response · Author response to Decision Letter 0]

5 Nov 2023

Authors’ response to reviewers

Manuscript title: 

Linkages between maternal experience of intimate partner violence and child nutrition outcomes: A rapid evidence assessment, PONE-D-23-02037

To the Editors, PLOSOne 

Thank you for the recognition of the contribution of our manuscript, “Linkages between maternal experience of intimate partner violence and child nutrition outcomes: A rapid evidence assessment.” In response to the reviewer and editor’s comments, several changes have been made, as detailed below. We feel that the suggestions have helped improve the manuscript. The comments are addressed point-by-point in turn below. 

1. Editor’s comments: 

We have reviewed and ensured that the manuscript meets at style requirements. 

2. Please identify your study as "systematic review" in the title of your manuscript.

The review is not a systematic review, rather a rapid evidence assessment. Therefore, we have retained the original title. 

3. PLOS requires an ORCID iD for the corresponding author in Editorial Manager on papers submitted after December 6th, 2016. 

We have added the ORCID iD for the corresponding author. 

Abstract: 

4. Line number: 52-58: The authors should provide a programmatic implications/policy with the special focus on low and middle-income countries.

We have added an additional programmatic implication and specified that it is for LMICs. The text now reads: “Programmatic implications include incorporation of GBV considerations into nutrition policies and programming and integrating GBV prevention and response into mother and child health and nutrition interventions in LMIC contexts.”

Introduction: 

5. Within the children which particular age group author focuses, its is missing in the introduction section.

We have added a definition of the age group of children, which is “defined as anyone from birth until age 18, as per international definitions.”

6. It would be good if author provide few literatures on gender wise differentiation on any child nutrition, IPV.

We have added consideration of gendered exposure and impacts of IPV. Regarding nutrition impacts, this is a gap that is identified in this systematic review and discussed in the Discussion section. We have added the following text: “Girls and boys may experience different patterns of direct and indirect exposure to IPV, however, the limited research exploring these differences is mixed [17, 18], and there are also gender differences in mental health impacts of childhood IPV exposure [19].”

7. Overall in whole introduction section need to provide country specific example it would and matched with low and middle-income countries.

We have added the following sentences in response to this comment: “For example, a study conducted in South Africa showed that emotional IPV was associated with lower language development, motor development and cognitive scores in children at age 2, while physical IPV was associated with lower motor scores [17], and in a study conducted in Kenya, maternal exposure to IPV was associated with poorer child behavioral outcomes [18].”

8. It’s better to cite recent systematic review study with the special focus on low and middle-income countries. Try to include systematic review which was published after 2020.

The systematic review cited is focused on LMICs. Unfortunately, there are no more recent systematic reviews that focus on prevalence of GBV in humanitarian settings, which is the issue we are discussing in that sentence. 

9. Line number: 94-98: Author provides the reference of first three line (line number: 94-96) also need to brief the sentences. No need to elaborate one single statement into different references. 

We have edited the sentence to shorten it. The sentence now reads: “Children – defined as anyone from birth until age 18, as per international definitions – can be indirectly exposed to IPV, which can include witnessing, being aware of, or secondarily affected by the presence of violence against a maternal caregiver in the household.”

10. The statement related to mental health among adolescents are very crude, Author should provide country specific case study or evidence also remove the word ‘Some children’ try to include better and clear words. 

We have edited this sentence. We do not understand the editor’s preference to remove the phrase “some children,” and feel that it accurately reflects the data and evidence in this field. The sentence now reads: “Evidence indicates that pre-school aged children who witness IPV in childhood may experience detrimental effects on their later mental, social and physical health during adolescence and adulthood.” 

Methods: 

11. I would suggest to use 'PICOTS' framework to summaries the eligibility criteria.

The PICOTS framework is not directly relevant to this systematic review as we do not have an I – intervention, that we are studying. However, we found the editor’s suggestion of using a framework to summarise eligibility criteria useful, and therefore, we have summarized the eligibility criteria using the PECO framework – Population, Exposure, Context, Outcomes. This section now reads: “The eligibility criteria were structured using the PECO framework – Population, Exposure, Context and Outcome [25]. The population was boys or girls under the age of 18 years. The exposure was any indirect exposures to IPV (in-utero and/ or household IPV exposures). IPV was defined as physical, sexual or psychological (including emotional and verbal) violence or controlling behaviors, perpetrated against a maternal caregiver. The context was a country in receipt of United Nations Central Emergency Response Funding during 2006-2021 [26], which we refer to as “humanitarian countries.” The outcomes were a) fetal growth (measured indirectly with birth weight), b) breastfeeding, including infant feeding practices, c) indicators of child growth (using anthropometry), and d) nutrient blood markers. The timepoints considered were during infancy, early childhood (including those outcomes specific to children under 2 or under 5 years), middle childhood and adolescence (after puberty or above age 10 years).” 

12. Line number: 157-160: It is good to provide the search terms in these three Medline, Embase, and Global Health electronic databases. This can be added as an Appendix A. Provide the search strategy with proper keywords, output and date of search. Overall appendix A need to revised and prepared as per the proper systematic review protocol.

We have updated Appendix A and it now includes the search terms for these three databases.

13. No need to provide ‘IPV’ definition in this section, either it is better to include in introduction section. 

The IPV definition is included in the introduction, and also here to specify that the eligibility criteria include any form of IPV. Some reviews focus only on specific types of IPV, so it is important to keep this here to adhere to reporting requirements for the review. 

14. Did the researchers use an application to support and track the screening process of the identified 3037 studies. If yes, please state what was used and how. Who is ‘one reviewer’, ‘two reviewers’, ‘third independent reviewer’? It would be better to provide the short indication name of the author after using these words.

We have indicated that we used Covidence and the initials of reviewers for clarity. This section now reads: “All records identified through the database searches were downloaded to Covidence, a systematic review software. Screening occurred in two stages: (i) title and abstract; and (ii) full text review. Title/ abstracts were each screened by one reviewer [one of SBC, MOG, LV or SRM]. During the full text screening, sources were assessed independently by two reviewers vis-à-vis the inclusion/ exclusion criteria [two of SBC, MOG, LV or SRM]. A third independent reviewer resolved any discord between the first two reviewers.” 

15. It would be clear, if researcher prepare the conceptual framework and include in the methods section.

In our experience conducting systematic reviews, scoping reviews and rapid evidence assessments, a conceptual framework is not required or usually included. This review was not conducted according to a specific conceptual framework, therefore we have not added one.

16. How was the data extracted and validated between the researchers.

We have added the following sentence to explain this: “The data extraction was completed by a single reviewer [one of SBC or SRM] and checked for consistency and accuracy by a second reviewer [MOG or SRM].”

17. How researcher was used four quality assessment tools in a single study? How did you validate your quality assessment based on using different questions? How did you calculate quality score? Overall, this section needs to revised and write very carefully and provide full explanation of how researcher uses these four different quality assessment tools? Is there any previous systematic review or rapid review that used same method of quality assessment if yes provide the references? 

We have added more detail on the specific items used in the quality assessment tool and the scales that each items was drawn from. We have found several systematic reviews that have used combined quality assessment tools in the way that we implemented it, and have cited one for the reader’s reference. We have added the following detail: “Specifically, the items whether the research objective was clearly stated, valid measurement of exposure (IPV) and outcome (nutrition) variables, and inclusion of relevant confounders were drawn from the NIH instrument, external validity (representativeness of sampling) and internal validity (non-response rates) items were drawn from the NOS instrument, an item about inclusion and exclusion criteria is drawn from the AXIS tool, and an item focused on sampling strategy is drawn from MMAT. This combination of items best fit the review objectives and included studies; previous systematic reviews have similarly combined quality assessment instruments [for example, 34].”

18. Author should attach quality assessment of each study and their score in a single file and attach in a supplementary file.

We have included this as a supplementary file. 

19. Risk of bias also missing in the method as well as result section, try to incorporate, in the revised manuscript.

The quality assessment tool that we implemented reflects risk of bias, therefore the discussion that is currently included in the results section is sufficient for the purposes of this rapid evidence assessment. 

Results

20. Line number: 204-205: S2 Table. Study Results should be categorized into four different categorise, namely: a) Fetal growth, b) Infant feeding, c) Child growth and d) Nutrient blood markers. 

We have revised S2 Table in line with this comment. 

Discussion 

21. Replace the word ‘room’ with other synonym word

We have replaced the word room with scope. 

22. Line number: 469-472: For this particular line “the studies suggest that maternal experience of IPV affects several growth indicators in preschool children, however, the literature is extremely limited in its examination of time intervals that can fully inform our understanding of any potential underlying mechanism” provide the reference.

By “the studies” and “literature” in this sentence, we are referring to the included studies in this review, all of which are referenced in the review. We have edited the sentence to clarify this: “the studies included in this review suggest that maternal experience of IPV affects several growth indicators in preschool children, however, the literature is extremely limited in its examination of time intervals that can fully inform our understanding of any potential underlying mechanism.” 

23. For this particular line “In contrast, there were seven out of twelve studies that found a positive association between maternal lifetime experience of any type of IPV and LBW” provide the reference.

We have added the references to these studies. 

24. Need to reframe this line “Regarding the association between a mother's experience of IPV and breastfeeding, our findings indicate a significant impact.”

We have rephrased the sentence to read: “Our findings find a significant association between mother's experience of IPV and breastfeeding.” 

25. If this line is indicating the limitation then it should be under the limitation section.

This is a limitation of the evidence-base that we identified, not a limitation of the systematic review itself. Therefore, we have retained its location in the discussion section. 

26. If this line is indicating the limitation then it should be under the limitation section. 

As per the previous response, this sentence indicates a limitation of the data analyses used in the included studies, not a limitation of this systematic review. 

27. In the discussion section, uniformity is missing, try to accumulate your finding based on your key outcome (Fetal growth, Infant feeding, Child growth, Nutrient blood markers). Also, I observed that between the discussion few line are highlighting the limitation and suggestion try to remove these lines and add those lines in the proper order.

We have restructured the Discussion section to reflect the order suggested by the editor, so the Discussion now focuses first on fetal growth, then feeding, then child growth and finally nutrient blood markers. As noted above, the lines about limitations throughout the discussion section refer to limitations in the evidence-base that were identified in the systematic review, not of the systematic review itself. 

28. Try to add ew lines on strength of this study except only writing the limitation section.

We appreciate this comment, and have added the following sentence in response: “Despite these limitations, this review represents an important update to prior evidence synthesis of the associations between maternal IPV and child growth and nutrition. By capturing the past few years of research, which has seen an enormous growth in published analyses, represents a comprehensive and up-to-date synthesis of this literature.”

29. The authors should provide a programmatic implications/policy with the special focus on low and middle-income countries. 

We have added the following in response to this comment: “Approaches including targeted IPV screening of women whose children are undergoing treatment for malnutrition through supplementary feeding programs, training pre-natal health care workers and lactation specialists to identify signs of IPV amongst women, and strengthening referral networks between health and nutrition services and IPV response services are all key actions that can be explored in LMICs.” 

Reviewer 1:

1. Define LMIC.

We have added a definition of LMIC the first time that it is mentioned. 

2. Why to include adolescents?

Adolescents fall under the definition of children, and the review sought to include all relevant data on children. 

3. Please elaborate line number 159-160. 

We have added a sentence elaborating on the search strategy: “In brief, the search terms included the following fields: i) intimate partner violence, ii) nutritional outcomes, and iii) quantitative study design, with specific terms and MeSH headings tailored to each database.” 

4. Methodology adopted to plot Harvest plot should be explained. Perhaps, after data synthesis (line no. 191- 194), the authors can add a section on the methodology adopted in the article.

Thank you for this comment, we agree that description of the harvest plot would be useful for the reader. We have added the following sentence: “Data synthesis also included developing harvest plots to visually represent the associations identified in studies; the method was adapted to the purposes of this review, whereby associations were not weighted by study quality [34, 35].”

---

## [Decision Letter · Decision Letter 1]

4 Jan 2024

PONE-D-23-02037R1Linkages between maternal experience of intimate partner violence and child nutrition outcomes: A rapid evidence assessmentPLOS ONE

Dear Dr. Meyer

Thank you for submitting your manuscript to PLOS ONE. After careful consideration, we feel that it has merit but does not fully meet PLOS ONE’s publication criteria as it currently stands. Therefore, we invite you to submit a revised version of the manuscript that addresses the points raised during the review process.

We look forward to receiving your revised manuscript.

Kind regards,

Pradeep Kumar, Ph.D.

Academic Editor

PLOS ONE

Journal Requirements:

Additional Editor Comments:

Reviewers' comments:

Reviewer #2: All comments have been addressed.

2. Is the manuscript technically sound, and do the data support the conclusions?

Reviewer #2: Yes

3. Has the statistical analysis been performed appropriately and rigorously? 

Reviewer #2: Yes

4. Have the authors made all data underlying the findings in their manuscript fully available?

Reviewer #2: Yes

5. Is the manuscript presented in an intelligible fashion and written in standard English?

Reviewer #2: Yes

6. Review Comments to the Author

Reviewer #2: The revisions that author(s) made to the manuscript effectively address all the comments/suggestions. I decide to accept the manuscript with minor acceptance.

Comment 1.

Line Number: 169 – 184 (PECO framework): convert into tabulation form.

Comment 2.

Line Number 195 -197: This line will start after the sentences of data extraction. Start from data extraction sentences after completion then author writes the quality assessment sentences.

7. PLOS authors have the option to publish the peer review history of their article (what does this mean?). If published, this will include your full peer review and any attached files.

Reviewer #2: No

---

## [Editor Report · Decision Letter 2]

24 Jan 2024

Linkages between maternal experience of intimate partner violence and child nutrition outcomes: A rapid evidence assessment

PONE-D-23-02037R2

Dear Dr. Meyer,

We’re pleased to inform you that your manuscript has been judged scientifically suitable for publication and will be formally accepted for publication once it meets all outstanding technical requirements.

Kind regards,

Pradeep Kumar, Ph.D.

Academic Editor

PLOS ONE

---

## [Editor Report · Acceptance letter]

6 Mar 2024

PONE-D-23-02037R2 

PLOS ONE

Dear Dr. Meyer, 

I'm pleased to inform you that your manuscript has been deemed suitable for publication in PLOS ONE. Congratulations! Your manuscript is now being handed over to our production team.

Kind regards, 

on behalf of

Dr. Pradeep Kumar 

Academic Editor

PLOS ONE